# Fecal Calprotectin Concentrations in Cats with Chronic Enteropathies

**DOI:** 10.3390/vetsci10070419

**Published:** 2023-06-28

**Authors:** Denise S. Riggers, Panagiotis G. Xenoulis, Dimitra A. Karra, Lena L. Enderle, Gabor Köller, Denny Böttcher, Joerg M. Steiner, Romy M. Heilmann

**Affiliations:** 1Department for Small Animals, College of Veterinary Medicine, University of Leipzig, 04103 Leipzig, Germany; lena.enderle@kleintierklinik.uni-leipzig.de (L.L.E.); romy.heilmann@kleintierklinik.uni-leipzig.de (R.M.H.); 2Clinic of Medicine, Faculty of Veterinary Science, University of Thessaly, Trikalon 224, 43100 Karditsa, Greece; pxenoulis@gmail.com (P.G.X.); dim0karra@gmail.com (D.A.K.); 3Department for Large Animals, College of Veterinary Medicine, University of Leipzig, 04103 Leipzig, Germany; koeller@vetmed.uni-leipzig.de; 4Institute of Veterinary Pathology, College of Veterinary Medicine, University of Leipzig, 04103 Leipzig, Germany; denny.boettcher@vetmed.uni-leipzig.de; 5Gastrointestinal Laboratory, School of Veterinary Medicine and Biomedical Sciences, Texas A&M University, College Station, TX 77843-4474, USA; jsteiner@cvm.tamu.edu

**Keywords:** small cell lymphoma, food-responsive enteropathy, steroid-responsive enteropathy, inflammatory bowel disease, gastrointestinal diet, hydrolyzed diet, feline chronic enteropathy, biomarker

## Abstract

**Simple Summary:**

Intestinal diseases in cats are challenging to diagnose, particularly when intestinal inflammation is to be distinguished from lymphoma, a neoplastic condition. In this study, levels of the protein complex known as calprotectin in fecal samples are investigated in both diseases and in comparison to healthy controls. The authors also investigated potential correlations between fecal calprotectin levels and clinical severity or intestinal microscopically visible changes of these diseases. There were no differences found between cats with lymphoma and inflammation in the intestines or cats with other diseases of the intestines, but calprotectin levels were higher in fecal samples of cats with intestinal diseases compared to healthy cats and cats with diseases located elsewhere in the body. This may indicate that calprotectin plays a role in gastrointestinal lymphoma as well as gastrointestinal inflammation and that these two diseases can not be separated by fecal calprotectin levels. While calprotectin may not be suitable as a marker to differentiate different chronic intestinal diseases, it can distinguish intestinal diseases from other diseases with overlapping clinical signs and from health. Further insights into the role of calprotectin will help better understand the disease pathogenesis and discover novel treatment avenues.

**Abstract:**

Diagnosis of feline chronic inflammatory enteropathies (CIE) and the differentiation from small cell intestinal lymphoma (SCL) can be challenging. Intestinally expressed calprotectin (S100A8/A9 protein complex) appears to be part of the complex pathogenesis of feline chronic enteropathies (FCE). Fecal calprotectin is a non-invasive biomarker for intestinal inflammation in humans and dogs but has not yet been evaluated in cats. We hypothesized that fecal calprotectin (fCal) concentrations are increased in FCE, correlate with clinical and/or histologic disease severity, and distinguish cases of CIE from SCL. This case–control study included fecal samples and patient data from cats with CIE (*n* = 34), SCL (*n* = 17), other gastrointestinal (GI) diseases (*n* = 16), and cats with no clinical signs of GI disease (*n* = 32). fCal concentrations were measured using the immunoturbidimetric fCal turbo assay (Bühlmann Laboratories). Compared to healthy cats, fCal concentrations were significantly increased in CIE, SCL, and other diseases (all *p* < 0.0001), but were not different between these three groups (all *p* > 0.05), or between cats with extra-GI diseases and healthy controls. These findings suggest that fCal may have utility as a clinical biomarker for FCE but not for intestinal disease differentiation. It further supports the role of calprotectin in the pathogenesis of the spectrum of FCE, which includes CIE and SCL.

## 1. Introduction

Feline chronic enteropathy (FCE) is a common disease of particularly elderly cats and its prevalence has increased over the last years [1,2]. Compared to the number of studies in dogs on chronic enteropathies, much less is known about FCE [3,4,5]. Currently, the definition of FCE includes the presence of clinical gastrointestinal (GI) signs for ≥3 weeks after the exclusion of extra-GI conditions (e.g., diabetes mellitus, hyperthyroidism) or other GI disorders (e.g., helminthic infections) [3,6]. Small-cell alimentary lymphoma (SCL) and chronic inflammatory enteropathy (CIE) [6] account for most cases of FCE.

Although the etiopathogenesis of CIE remains incompletely understood, it appears to result from a complex interplay of a genetic predisposition, dysregulated immune response, and environmental factors [4,6]. In most studies, the subclassification of CIE is based on the response to treatment into immunosuppressant-responsive enteropathy (IRE) and food-responsive enteropathy (FRE). These two entities cannot necessarily be differentiated based on histology [6,7]. Probiotic-responsiveness, as discussed in dogs [8], has not yet been demonstrated in cats but may also play a role. Given the frequent coexistence of neoplastic and inflammatory lesions and/or a previous diagnosis of CIE in cats diagnosed with intestinal lymphoma, particularly intestinal SCL, progression of CIE to SCL over months to years has been suspected [9,10,11].

SCL accounts for approximately 75% of all GI lymphoma cases in cats. Histologically defined by populations of small lymphocytes that are well differentiated and have low mitotic rates, SCL usually is associated with a slow clinical progression [12].

Currently, it can be difficult to subclassify feline CIE into the subgroups IRE and FRE and differentiate it from SCL or other diffuse infiltrative neoplasia [6]. SCL and CIE largely overlap in terms of patient characteristics (i.e., age, breed), clinical signs (i.e., weight loss, vomiting, anorexia, diarrhea), and noninvasive diagnostic findings (i.e., abdominal ultrasonography, routine blood work) [5,6,13,14,15,16,17,18,19,20].

The current treatment for IRE and SCL cases in cats is immunosuppression and/or chemotherapy [1,4,6], which can have significant side effects. For the small animal practitioner, more targeted treatment options tailored to specific pathways would be desirable for CIE cases. Therefore, further investigation of the pathogenesis of FCE, particularly the inflammatory pathways, will lead to an improvement in the general understanding of these diseases and potentially open new avenues for individualized therapeutic interventions.

The S100A8/A9 protein complex (calprotectin) is a DAMP (damage-associated molecular pattern) molecule of the innate immune response. In dogs with intestinal inflammatory diseases, fecal calprotectin is a non-invasive biomarker [21,22,23,24,25]. Calprotectin is expressed primarily by activated macrophages (MΦ) and neutrophils, but can also be induced in epithelial cells [26,27,28] and acts as a ligand for Toll-like receptor (TLR)-4 [29]. In human medicine, fecal calprotectin (fCal) is used routinely for the diagnosis and treatment monitoring of patients with inflammatory bowel disease (IBD) [30].

In dogs with CIE, levels of fCal have been shown to be increased and correlate with disease subclassification, severity of clinical signs, histologic changes, and endoscopic lesions [22,23,24]. fCal is a relatively inexpensive noninvasive biomarker that can be used in the diagnosis, subclassification, and potentially the prediction of response to treatment in human IBD and canine CIE [1,22,24,28,31,32,33,34,35,36,37,38,39]. fCal concentrations are also more specific for GI disease compared to serum calprotectin concentrations [23,24,40]. Furthermore, fCal appears to resist degradation by intestinal enzymes and bacteria, with fCal concentrations shown to be stable in fecal samples for up to one week at room temperature [24,37,38,41].

The fCal^®^ turbo assay (Bühlmann Laboratories, Schönenbuch, Switzerland) is a polyclonal antibody particle-enhanced turbidimetric immunoassay (PETIA) for the measurement of calprotectin in fecal extracts [42]. This assay has recently been validated for the use with samples from cats [43].

Preliminary data suggest that both intestinal mucosal calprotectin expression [20] and fCal concentrations [39] are significantly increased in chronic GI inflammation in cats, but the diagnostic and/or prognostic value of fCal measurement in FCE patients remains unknown. Clinical disease severity, hypoalbuminemia, hyperglobulinemia, and some endoscopic and individual histologic (inflammatory) changes have been associated with mucosal S100/calgranulin expression [20]. Thus, these DAMP molecules and their signaling pathways could be attractive targets for disease-specific fecal biomarkers and potential new avenues to disease-specific therapeutic intervention in cats with FCE [20].

Our central hypothesis for the study was that fCal measurement has clinical utility in the diagnosis of FCE and/or the prediction of treatment response, and that fCal concentrations can distinguish cats with SCL, CIE, and healthy controls. Furthermore, we hypothesized that fCal concentrations correlate with the severity of clinical disease. Thus, the aim of our study was to investigate fCal concentration in fecal specimens from cats with CIE, SCL, and a group of healthy control cats.

## 2. Materials and Methods

### 2.1. Study Population and Routine Diagnostics

Fecal samples of 105 cats with suspected CIE were prospectively collected and complete medical records of the cats were reviewed (Figure 1). Ethics approval was not required to collect naturally passed fecal samples from cats for this study (German animal welfare regulations; Animal Welfare Act). Cats were selected for inclusion in the study based on (i) the presence of gastrointestinal (GI) signs for ≥3 weeks, (ii) not receiving any medication that may affect calgranulin expression and/or release (e.g., non-steroidal anti-inflammatory drugs or corticosteroids [44]) for at least 4 weeks before fecal specimen collection, and (iii) exclusion of extra-GI conditions.

To exclude the presence of other conditions and assess the overall patient health, a minimum database (serum biochemistry profile, complete blood cell count, fecal examination by flotation), *Giardia* spp. antigen-ELISA, and abdominal ultrasonography were performed. Sonographic findings considered as abnormal were an increased total GI wall thickness (duodenal/jejunal wall >2.5 mm and ileal wall >3.2 mm [45]), increased thickness of the muscularis layer (>0.3 mm for the duodenal muscularis layer, >0.4 mm for the jejunal muscularis layer, and >0.9 mm for the ileal muscularis layer), loss of GI wall layering, enlargement of mesenteric lymph nodes, and ascites [20,46]. Fecal scores [47] were determined for 14 cats (12 cats with CIE, 2 with SCL, and 10 GI controls).

If indicated, cobalamin, folate, fructosamine, total thyroxine (tT4), fPLI (serum feline specific pancreatic lipase, measured by Spec fPL, Idexx Laboratories, Westbrook, ME), and feline trypsin-like immunoreactivity (fTLI) concentrations, and retrovirus (FeLV/FIV) testing were performed. Some cats were also tested for *Tritrichomonas blagburni* (formerly *T. foetus*).

Cats with chronic GI signs that did not respond to dietary intervention and those with a strong suspicion for SCL underwent further diagnostic evaluation, including a GI endoscopy with collection of mucosal biopsies of the upper and, if indicated, also the lower GI tract.

Per GI section and animal, at least 5 endoscopic tissue biopsies were obtained. The tissue biopsies were evaluated by routine histology according to the criteria of the WSAVA Gastrointestinal Standardization grading system [48]. Morphologic lesions and inflammatory changes in the stomach, duodenum/proximal jejunum, ileum, and colon were assessed each on a 4-point scale (0 = normal, 1 = mild lesions, 2 = moderate lesions, and 3 = severe lesions) to calculate cumulative lesion scores (i.e., the sum of individual lesion scores).

If histological findings were not sufficient to differentiate SCL from lymphoplasmacytic enteritis, immunohistochemistry (IHC) for CD3 (T cell) and CD20 (B cell) was performed and interpreted based on the location and proportion of positively staining cells.

To assess the clinical disease severity, FCEAI scores [5] were retrospectively calculated for all cats at the time of diagnosis. The FCEAI scoring system evaluates the following parameters: attitude/activity, endoscopic lesions in the GI tract, serum ALT/ALP activity, phosphorous concentration, abnormal serum total protein (TP) concentration, and the presence of GI signs, which include diarrhea, vomiting, hyporexia, and/or weight loss. Based on their severity, clinical signs were graded from 0–3 (normal = 0, mild = 1, moderate = 2, severe = 3). All other variables were dichotomously scored as either 0 = normal or 1 = decreased (albumin, phosphorus) or increased (TP, ALT, ALP) [5]. The cumulative FCEAI scores were categorized as mild clinical disease (with a FCEAI score of 0–5), moderate clinical disease (with a FCEAI score of 6–12), or severe clinical disease (with a FCEAI score of 13–19).

By using a standardized questionnaire, patient follow-up information from the owners and referring and/or attending veterinarians were obtained.

Based on this data, the survival times and responses to treatment were determined, and cats with a diagnosis of CIE were subclassified as either IRE or FRE.

Cats with GI signs and differential diagnoses other than FCE were also included in the study as a disease-control group and were required to (i) have a final diagnosis established and (ii) not have been receiving any medication that may affect GI calgranulin expression and/or release at the time of fecal specimen collection (Figure 1).

Fecal samples were obtained from 32 healthy cats owned by staff members and students of the CVM-LU or presented for routine check-up evaluations and were used as a healthy control group. The owners of these cats completed a standardized study questionnaire to confirm that all control cats were free of any signs of GI disease and/or other conditions. All control cats underwent fecal examination by flotation.

### 2.2. Sample Collection and Processing

Fecal samples collected after natural defecation were either immediately placed into a specially designed sampling tube filled with the extraction buffer (Calex Cap; Bühlmann) yielding a final dilution of 1:500 or were stored frozen at −20 °C and then transferred into the Calex Cap device. The samples were collected from ≥5 different aliquots of each fecal sample [43]. All fecal extracts were stored refrigerated at 6 °C for up to 24 h after shaking incubation at room temperature (~23 °C) for 20 min, and were then transferred to storage at −20 °C until analysis. For further processing and analysis, the samples were defrosted, adjusted to room temperature, centrifuged at 1500× *g* for 5 min, and the supernatant was used for the fCal assay.

### 2.3. Fecal Calprotectin Measurement

Calprotectin was measured in all fecal extracts using the fCal turbo assay on a Roche Cobas 311 chemistry analyzer as previously validated for feline specimens [43]. Briefly, fecal extracts were incubated with proprietary reaction buffer and mixed with polystyrene nanoparticles that had been pre-coated with polyclonal anti-human calprotectin antibodies. Agglutination by binding of fCal increases sample turbidity and was measured at 546 nm and 800 nm, and the fCal concentration was determined by interpolation from the calibration curve [42,43]. The assay has a working range from 3–2000 μg/g for feline samples, and samples with a fCal concentration >2000 μg/g were re-assayed in a 1:1000 dilution.

### 2.4. Statistical Analysis

Commercial statistical software packages (JMP^®^ v.13, SAS Institute, Cary, NC, USA; GraphPad Prism v.9, Dotmatics, Boston, MA, USA) were used for all statistical analyses. For testing the normality of the data, a Shapiro–Wilk W test was used. To report summary statistics, counts (n) and percentages for categorical data and medians and interquartile ranges (IQR) for continuous data were used. Non-parametric two-group (Wilcoxon signed-rank test) or multiple-group comparisons (Kruskal–Wallis test followed by Dunn’s post-hoc test) were performed, and associations were tested using the likelihood ratio or Fisher’s exact test. A non-parametric Spearman correlation coefficient (ρ) was calculated to test for possible correlations between or among continuous variables. Statistical significance was defined as *p* < 0.05, and a Bonferroni correction was applied for multiple comparisons or associations if indicated.

## 3. Results

### 3.1. Patient Clinical Characteristics

A total of 51 cats with FCE, 34 with CIE, and 17 with SCL were included in the study. For the controls, 16 cats with other diseases (disease controls) and 32 healthy control cats were used. The disease control group of cats included feline patients with acute GI disease (*n* = 9), other chronic GI diseases (*n* = 2), and extra-GI disease (*n* = 5). Specifically, the acute GI disease group included cats diagnosed with acute gastroenteritis associated with intestinal dysbiosis (*n* = 2), acute GI signs associated with rehoming stress or infection (*n* = 2), acute gastroenteritis of unknown cause (*n* = 3), helminth infestation (*n* = 1), and parvovirus enteritis (*n* = 1). In the group of cats classified as other GI disease, one cat had a mesenchymal rectal neoplasm and the other cat an uncharacterized enteropathy with hyperthyroidism and potential CIE. The cats classified as extra-GI disease were diagnosed with hyperthyroidism (*n* = 3), diabetes mellitus (*n* = 1), or feline infectious peritonitis (FIP, *n* = 1).

The cats with SCL were significantly older than those cats diagnosed with CIE (*p* = 0.0014; Table 1), disease controls (*p* = 0.0437), or healthy controls (*p* < 0.0001). In addition, the cats with CIE were significantly older than the healthy controls (*p* = 0.0421). There were no statistically significant differences in the sex distribution, body condition scores, body weights, or any other patient characteristics (all *p* > 0.05) among these groups of cats. In all four groups, most animals were Domestic Shorthair cats. A negative retrovirus status was determined in all cats that underwent testing. Survival time after diagnosis and disease duration before presentation for diagnostic investigation did not differ among the groups (*p* = 0.1413), but survival time after diagnosis was significantly shorter in the cats diagnosed with SCL compared to the disease controls (*p* = 0.0048). *Giardia* spp. coproantigen was tested in 24 cats (11 cats with CIE that were all negative, 5 cats with SCL of which 3 were positive, and 8 disease controls with 2 being positive). The presence of *Tritrichomonas blagburni* (formerly identified as *T. foetus* was evaluated in 37 cats (21 cats with CIE, 11 cats with lymphoma, and 5 disease controls) which were all negative. All the cats underwent diagnostic imaging, with complete still images and/or video sequences of abdominal ultrasound available for review from 26 cats (13 cats with CIE, 2 cats with SCL, and 11 disease controls). There was no significant difference in the sonographic findings among the cats with SCL, CIE, and disease controls (all *p* > 0.05), except for a higher rate of increased intestinal wall thickness in the cats with CIE or SCL compared to disease controls (*p* = 0.0078). FCEAI scores in the cats with SCL were significantly higher compared to the CIE cases (*p* = 0.0056) and disease controls (*p* = 0.0050), with more severe vomiting and diarrhea in the SCL cases (Table 1). Hematochezia or melena was significantly more frequently seen in the cats with SCL than in those with CIE (*p* = 0.0441). The onset of clinical signs being triggered by stress was significantly more common in the disease controls than in the SCL cases (*p* = 0.0278), but no difference was seen between the CIE and SCL cases (*p* = 0.1333) or disease controls (*p* = 0.4667).

### 3.2. Clinicopathologic Evaluation

Cobalamin concentrations in serum were significantly lower in the SCL group of cats compared to the CIE cases (*p* = 0.0015). In addition to mild variations in serum albumin, globulin, and total tT4 concentrations among the groups of cats, serum BUN concentrations were significantly higher in the SCL cases than in the cats with CIE (*p* = 0.0107) or disease controls (*p* < 0.0001). Serum fTLI concentrations were also significantly higher in the cats with SCL compared to those with CIE (*p* = 0.0013). No differences among the groups of cats were detected for any other serum biochemistry parameter or the frequency of leukocytosis or anemia (all *p* > 0.05). Proteinuria (increased urine protein/creatinine ratio) was detected in one disease control case.

### 3.3. Histologic Examination

Endoscopy with biopsies was performed in 50 cats (esophagogastroduodenoscopy combined with ileocolonoscopy or colonoscopy in 46 and 4 cases). For ethical reasons, GI tissue biopsies were only available from some of the disease controls (two cats with endoscopy performed due to the clinical suspicion of FCE) and were not obtained from any of the cats in the healthy control group. Endoscopic lesions were detected in the proximal small intestine of almost all of the cats (86–100% in each disease group) and in the stomach of about half of the cats (43–50%), with no differences among the groups of cats. Linear hyperemic lesions were detected in seven cats with CIE, one cat with SCL, and one disease control cat.

In the CIE group, inflammatory lesions were detected in the stomach in 19 cats (76%), duodenum or proximal jejunum in 24 cats (96%), ileum in all 23 cats (100%), and colon in 21 cats (84%). In the cats with SCL, gastric biopsies revealed inflammation in 10/17 cases (59%), gastric SCL was detected in 7/17 cases (41%); duodenal or proximal jejunal inflammation in 2/17 cases (12%), whereas 15/17 cats (59%) were affected by duodenal/proximal jejunal SCL; and 14/14 cases had inflammation in the colon. SCL was diagnosed in ileal biopsies of 14/15 cats (93%). Immunohistochemistry for CD20 and/or CD3 was performed in 21 cats, 9 of which were classified as CIE and 12 cats diagnosed with GI-SCL. Helicobacter-like organisms (HLO) could be detected in gastric biopsies from five cats (four cats with CIE and one disease control).

### 3.4. Treatment Response

Follow-up data were available from 30 cats in the group with CIE (17 cats classified as FRE and 13 as IRE) and from the 17 SCL cases. In the CIE group, 17 cats (56%) achieved complete remission (CR), 11 cats (37%) achieved partial remission (PR), and 2 cats (7%) had no clinical response (NR). Of the 17 cats with FRE, 2 cats responded to a commercial limited-ingredient (novel monoprotein) diet, 11 cats to a commercially hydrolyzed diet (of which 3 cats had failed a prior dietary trial with a novel protein diet), and 4 cats to an easily digestible prescription GI diet. All 13 cats diagnosed with IRE had failed dietary trials (i.e., PR or NR) with a GI and monoprotein diet (1 cat), hydrolyzed protein diet (9 cats), or both sequentially (2 cats) and showed resolution of clinical signs (CR) or significant improvement under anti-inflammatory or immunosuppressive (oral prednisolone) medication. In one cat classified as IRE, dietary intervention could not be performed due to food aversion. Additional treatments included antiemetics (3 cats), prebiotics and/or probiotics (4 cats), cobalamin supplementation (19 cats), appetite stimulants (2 cats), and gastroprotection (3 cats). Remission rates were higher in cats with FRE (75%) than in cats with IRE (36%), but the difference did not reach statistical significance (*p* = 0.0608).

All cats with GI-SCL had improved clinical signs on chemotherapy. Chlorambucil combined with prednisolone was the most frequently used chemotherapy protocol (13 cats), 2 cats underwent prednisolone monotherapy, and 2 cats received only symptomatic treatment. Additional treatments in IRE cats included antiemetics (four cats), gastroprotective drugs (two cats), appetite stimulants (two cats), analgesics (one cat), and an antimicrobial (one cat). Diet was changed in two cats (hydrolyzed protein and GI diet each in one cat). Response to treatment and survival time after diagnosis did not differ significantly between the cats with SCL and the cats with IRE (*p* = 0.0618 and *p* = 0.6655).

### 3.5. Fecal Calprotectin Concentration

Fecal calprotectin (fCal) concentrations ranged from 3–5487 µg/g (median: 19 µg/g) in all cats, with the highest measurement detected in a cat with rectal neoplasia. Compared to the healthy control cats, fCal concentrations were significantly higher in the cats with CIE, SCL, and in the cats with other diseases (all *p* < 0.0001; Figure 2). No significant differences were detected among these three disease groups (CIE, SCL, and other diseases; all *p* > 0.05; Table 1). Fecal calprotectin concentrations did also not differ among the cats classified as FRE, IRE, or SCL (all *p* > 0.05; Figure 3).

Fecal calprotectin concentrations were significantly higher in the cats with acute GI disease compared to the controls (*p* > 0.0001) but showed no difference compared to the cats with CIE (*p* = 0.6073) or SCL (*p* = 0.3282). Concentrations of fCal were lowest in the cats with extra-GI disease and did not differ from the healthy controls (*p* = 0.6919). Fecal calprotectin concentrations were also higher in the cats with CIE compared to those with extra-GI disease (Figure 3), but the difference did not reach statistical significance (*p* = 0.0927).

### 3.6. Association of Patient Characteristics with Fecal Calprotectin Concentrations

In the cats with FCE (CIE or SCL), fCal concentrations were not correlated with the FCEAI score or any of its individual parameters, Waltham fecal score, patient age, body weight, or body condition score (all *p* > 0.05). In cats with CIE, higher fCal concentrations were moderately correlated with higher serum globulin (ρ = 0.48, *p* = 0.0104) and lower serum albumin (ρ =−0.45, *p* = 0.0165), BUN (ρ = −0.46, *p* = 0.0162), cobalamin (ρ = −0.40, *p* = 0.0335), and folate (ρ = −0.42, *p* = 0.03154) concentrations. Fecal calprotectin concentrations were also linked to the presence and severity of several morphologic and inflammatory criteria on small intestinal mucosal biopsies from cats with CIE (Table 2). In cats with SCL, higher fCal concentrations were linked to lower serum ALT activities (ρ = −0.68, *p* = 0.0058), serum phosphorus (ρ = −0.56, *p* = 0.0317), and folate concentrations (ρ = −0.56, *p* = 0.0284). Fecal calprotectin concentrations were unaffected by fecal scores (ρ = 0.08, *p* = 0.7883).

In the cats of the CIE or SCL group, higher fCal concentrations were seen in the cats with anemia (median: 226 µg/g, IQR: 63–519 µg/g vs. median: 28 µg/g, IQR: 13–110 µg/g; *p* = 0.0013), which remained significant for the CIE group (median: 258 µg/g, IQR: 45–742 µg/g vs. median: 28 µg/g, IQR: 83–145 µg/g; *p* = 0.0125) but not the SCL group (median: 27 µg/g, IQR: 14–72 µg/g vs. median: 186 µg/g, IQR: 69–249 µg/g; *p* = 0.0519). Leukocytosis (*p* = 0.2453) and the presence of melena or hematochezia (*p* = 0.4306) were not associated with differential fCal concentrations.

Fecal calprotectin concentrations were also higher with the presence of HLO in gastric biopsies (median: 840 µg/g, IQR: 519–1246 µg/g vs. median: 30 µg/g, IQR: 5–100 µg/g; *p* = 0.0028), but none of the abdominal ultrasonographic or endoscopic parameters were significantly associated with differential fCal concentrations (all *p* > 0.05). Partial (PR) or non-responders (NR) in the CIE group of cats, specifically IRE, had higher fCal concentrations than the CIE/IRE cats with CR, but the difference did not reach statistical significance (Figure 4).

## 4. Discussion

In this study, fecal concentrations of calprotectin (fCal) were evaluated in cats with FCE, CIE, or SCL, and were compared with each other, a group of healthy control cats, and a disease control group comprised of cats with other GI or extra-GI diseases. This is the first investigation into fCal concentrations in cats with SCL and CIE.

The results of this study support our hypothesis that calprotectin might aid in the diagnosis of FCE. Significant differences between fCal concentrations in healthy cats and the group of extra-GI diseases compared to the other disease groups suggests that fCal has potential as a marker for FCE in cats with chronic GI signs and for CIE if acute disease and GI neoplasia have been excluded (pre-test probability) [25]. However, levels of fCal did not differentiate between CIE, SCL, and the GI control group. All disease groups included cats with fCal levels within the reference interval (<64 µg/g) [43]. Thus, as a diagnostic test, fCal will always have to be interpreted together with other diagnostic criteria.

Studies on dogs with acute hemorrhagic diarrhea syndrome (AHDS) showed that fCal concentrations increase during the acute disease phase and decline within 2–3 days of patient stabilization [49]. The cat with the highest fCal concentration in our study was a cat in the disease control group that was diagnosed with a mesenchymal (spindle cell) rectal neoplasm, and did not show hematochezia or melena. Studies in human patients have also shown fCal concentrations to be markedly increased in patients with colorectal neoplasia [50,51], thus fCal might also have potential as a GI tumor marker in cats. However, with only one cat affected or confirmed with non-SCL GI neoplasia in this study, further investigation of fCal in feline patients with GI neoplasms is warranted. We also acknowledge the small group size of the subgroups included as disease controls as this comparison was a secondary aim of the study. Thus, further investigations of fCal in other GI diseases is warranted.

The fact that we found no difference in the fCal concentrations between cats with CIE and SCL might support the theory that both conditions likely present different stages along a spectrum or continuum of one disease process, instead of being distinct diseases. This theory proposes that CIE—initiated by a chronic stimulus such as bacterial and/or food antigens or (unknown) viral pathogens—progresses to SCL over time [6,13,18,20,46,52].

A general limitation in all studies on feline CIE is the current definition of a “GI-healthy phenotype” in cats. For histologic evaluation, the WSAVA criteria were used in this study, but cats defined as “normal” that were used for establishing the WSAVA guidelines were young, specific pathogen-free (SPF) colony cats, which does not appear to represent an adequate control for the typical feline GI patient in clinical practice [48,52]. In addition, a recent investigation of 20 clinically healthy middle-aged cats [53] showed abnormal histologic findings (based on WSAVA criteria) in all cases, but even after approximately 2 years, only 15% of these cats developed GI signs [53]. However, our study design with long follow-up times and detailed follow-up information allowed for a reliable classification into FRE vs. IRE (vs. SCL) in a large number of cats.

Furthermore, differentiation of CIE and SCL based on histology can also be challenging. While as many diagnostic parameters as possible were included and the diagnosis was confirmed by CD3-/CD20-staining [6,54] and long follow-up times, the possibility of “hidden” SCL in cats of the CIE group and/or undiagnosed CIE or SCL in the control groups remains, particularly in cats not undergoing endoscopy with biopsy, if ileoscopy with ileal biopsies was not included in the endoscopic evaluation, or if present in segments of the GI tract that are outside the reach of the endoscope.

Compared to fCal concentrations in a group of dogs with CIE (median: 92 μg/g; range: 0–638 μg/g; [24]) and fCal concentrations in human patients with ulcerative colitis (mean ± standard deviation: 132 ± 97 µg/g [55] or >130 µg/g [56]) or inflammatory bowel disease (IBD; mean ± standard deviation: 652.8 ± 799.7 µg/g or >50 µg/g [57]), fCal concentrations were lower in the CIE group of cats. A possible explanation for generally lower fCal concentrations in cats compared to humans is the predominance of a lymphoplasmacytic infiltrate in feline CIE [58] as opposed to neutrophilic granulocytes that express calprotectin and predominate in human IBD [59]. However, it should also be noted that immunoassays are not true analytical assays and thus concentrations between species are not always comparable.

Our second hypothesis of fCal concentrations correlating with clinical disease severity (determined as FCEAI score [5]) could not be confirmed, which is consistent with our previous investigation of mucosal calgranulin expression in cats, showing only a moderate correlation between mucosal S100/calgranulin-positive cell counts and the severity of diarrhea in cats with SCL but no correlation with FCEAI scores [20]. Due to our study design and ethical constraints, not all cats underwent GI endoscopy, in some cases the patient medical data were incomplete, and individual case management, diagnostic evaluation, and follow-up times and intervals varied. Thus, single FCEAI parameters were not available for some cats (endoscopy in 7 cats of the CIE group and 13 cats of the GI control group; total protein concentration in 2 cats each with CIE and SCL and 3 GI control cats; ALT/ALP activity in 2 each of the GI controls and CIE cats and 1 SCL cat; and serum phosphorus concentration in 6 cats with CIE, 1 cat with SCL, and 8 GI controls) for FCEAI calculation. Calculation and analysis of FCEAI scores was considered reasonable given that this is, to the authors’ knowledge, also the first study to include FCEAI scores for GI control cats and a categorical classification of disease severity (i.e., mild, moderate, severe) based on FCEAI scores. Still, assessment of some FCEAI parameters (i.e., vomiting, diarrhea, weight loss, appetite, and especially activity), relies on observations by the owners and/or attending veterinarians, barring the risk of interpretive bias. Particularly in cats with chronic diseases, subtle clinical signs (e.g., mildly reduced activity level) might be overlooked or underestimated by the owners, and diarrhea and/or vomiting can be challenging to evaluate in cats with access to the outdoors.

Though not statistically significant, there was a trend of fCal concentrations to be higher in PR/NR cats with CIE compared to those cats reaching CR. Similar findings have been reported in human IBD patients [60,61] and also in dogs with CIE [22], suggesting a potential of fCal as a marker for the response to treatment in feline patients with chronic enteropathies. Longitudinal studies are needed and are currently underway to further investigate this hypothesis. The criterion of remission was based on non-invasive diagnostics and clinical signs in this study because no cat underwent repeated endoscopy for ethical reasons. This presents a disadvantage of using the FCEAI score for patient monitoring, and a simplified clinical scoring system for FCE to allow reevaluation of the cat without repeated endoscopy as the scoring system used for dogs [62,63] would be highly desirable.

We found fCal concentrations to be positively correlated with villus stunting, but inversely correlated with crypt distension in the duodenum/proximal jejunum. A similar trend was not detected in our previous study on intestinal mucosal calprotectin expression, but further studies comparing fCal concentrations with the corresponding mucosal expression of calprotectin are warranted.

The cats with SCL were older than the cats with CIE, with an overlap in the age ranges, as also described in former studies [20,53]. Domestic shorthair cats being overrepresented in all groups of cats should not be overinterpreted with this breed being most frequently seen in clinical practice throughout different geographic regions. Most clinical parameters also did not differ between the cats with CIE and SCL which is consistent with the current literature [6,46,64] and is another reason for the differentiation between CIE and SCL being challenging. The only significant difference among the sonographic criteria was a higher rate of increased intestinal wall thickness in the cats with FCE (CIE or SCL) compared to the disease controls, without differences between CIE and SCL, which also confirms the findings of others [46,65,66].

Serum BUN concentrations were significantly higher in the SCL cases compared to the cats in the CIE group and disease controls, which might reflect the increased risk of chronic kidney disease (CKD) in older cats [67], given the age difference between the groups. It is also possible that more cats in the SCL group were dehydrated and had prerenal azotemia. However, none of the cats in the study had a serum symmetric dimethylarginine (SDMA) and/or creatinine concentration indicating CKD IRIS stage II or higher. Another explanation for the higher BUN concentrations in the cats with SCL may be a higher rate or risk of occult GI bleeding in this group. Interestingly, the presence of hematochezia or melena (both interpreted together because of *n* = 1 cats with melena) had no effect on fCal concentrations. However, only macroscopic findings in stool samples were evaluated without further testing for occult fecal blood.

The cats with SCL showing significantly lower serum cobalamin concentrations confirms the results of a previous report [46]. However, despite ruling out an effect of prior cobalamin supplementation, the patients’ diets prior to diagnostic evaluation and sample collection varied, with most cats fed different commercial non-prescription diets with likely different levels of cobalamin fortification.

Serum fTLI concentrations were also significantly higher in the cats with SCL than in the cats with CIE. While we can only speculate as to the cause of this finding, possible explanations are a higher rate of pancreatic activation (e.g., mild/subclinical, chronic pancreatitis) or higher rate of CKD (as serum fTLI can increase with reduced renal excretion) in the SCL group of cats. Further studies in cats with SCL should shed more light on this phenomenon.

Response to treatment and minimum survival time after diagnosis did not differ significantly between the cats with SCL and cats with CIE, but the GI controls had significantly higher minimum survival times than the FCE cases. This must not be overinterpreted as most of the GI controls included in the study had a longer follow-up period than FCE cases of which a significant number could only be monitored for 3 months. With the intent to evaluate the potential utility of fCal concentrations to diagnose FCE and predict the response to treatment, a 3-month follow-up period was deemed reasonable and sufficient to assess short- and intermediate-term responses. Longer follow-up periods that allow to assess for potential relationships between fCal concentrations and survival time will require further study. Of note, two cats in the CIE group, three cats with SCL, and two GI-disease controls died or were euthanized for GI-related reasons during the follow-up period, and two GI-disease controls had a negative outcome due to extra-GI causes.

We also evaluated the possibility of an effect of stress as a trigger for clinical signs in the feline population of this study. Owner-perceived stressors for the individual cat were significantly more common in disease controls compared to SCL cases, with no difference between CIE and SCL or disease controls. This information could only be obtained by carefully questioning the owners, without the possibility of an objective validation of their subjective opinion. Previous studies have demonstrated that signs of stress in cats are very often not recognized or underestimated by owners and veterinarians alike, and many cats suffer from chronic distress, which can cause or exacerbate health problems by suppressing the immune system, as reported for feline idiopathic cystitis, atopic dermatitis, or acral lick dermatitis [68,69,70,71,72]. Associations between stress and GI conditions have been described in small animals [73,74], potentially caused or potentiated by an altered integrity of the intestinal barrier and, consequently, increased intestinal permeability and localized inflammatory reaction [75]. Therefore, potential stress factors (e.g., insufficient resources in the cat’s environment, incompatible social partners and other companion animals in the same household, lack of opportunities to satisfy playing and hunting behaviors) should be emphasized in the patient history when evaluating FCE patients, and should also be addressed as part of the therapeutic plan. There are several options to reduce stress in cats, including habituation, improved handling, environmental enrichment and resources, creation of safe areas and anxiolytic treatment such as psychoactive drugs, synthetic facial pheromones, aromatherapy, or nutritional supplements [72]. Supplementation of the essential amino acid tryptophan may also be recommended in cats with CIE [76] and to support coping with stressful situations [77,78]. In dogs and mice, reduced tryptophan levels correlate with the severity of GI disease [79,80,81,82], and a study on dogs showed decreased serum tryptophan concentration with protein-losing enteropathy as a subclass of CIE [83]. Tryptophan also decreases intestinal permeability and expression of pro-inflammatory cytokines [84,85]. Thus, evaluation of tryptophan metabolism and supplementation in FCE appears to be an interesting avenue for further studies. Whether cats with GI disease may benefit from other stress-reducing nutritional supplements, such as α-casozepine [78], also warrants further research.

Some of the cats in our study lacked an initial response to a dietary approach with hydrolyzed diet due to lacking acceptance of the diet but clinically improved on a novel monoprotein diet. Thus, it may be prudent to aim for at least three different strict elimination diet trials before reaching for more invasive diagnostics and/or immunosuppressive treatment in a cat that is clinically stable and refuses to eat, for example, a hypoallergenic diet. Cats with FRE have better long-time outcomes and chances of remission than those with IRE [20]. Explaining the importance of a rational dietary approach to owners of affected cats is also vital and will increase acceptance and owner compliance. This includes a stepwise change from the previous to the new diet and tips on making the food more attractive or palatable (e.g., warming, change in consistency by adding water or non-soluble fibers, adding prebiotics/probiotics to food). In line with this, one cat in the FRE group received additional treatment with the fungus *Coriolus versicolor*. A study in mice showed this fungus to suppress IBD development by inhibiting STAT1 and STAT6 as well as IFN-γ and IL-4 expression [86]. To the authors’ knowledge, no studies on its effect in feline patients exist, but this one cat appeared to respond, and the clinical signs worsened with every attempt to discontinue this supplement. Thus, further studies should evaluate the potential benefit of phytotherapeutics in cats with FCE.

We acknowledge some limitations of our study. First, the overall group size was small, thus a type II error for finding no difference or association cannot be excluded. Particularly, the GI-disease control group might have benefited from a larger group size, which warrants further studies with larger groups to further evaluate fCal levels in different conditions. Furthermore, due to the study design, not all cats underwent GI endoscopy; in some cases, patient medical data were incomplete, and there was some variation in individual case management, diagnostic evaluation, and times of follow-up evaluations. Lastly, fCal concentrations could not be directly compared to mucosal calprotectin expression in this study.

## 5. Conclusions

The findings of our study show that fCal concentrations are higher in FCE cats compared to healthy controls. However, fCal concentrations cannot differentiate between CIE and SCL, which may present different stages along the spectrum of FCE. Cases of extra-intestinal disease may be distinguished by fCal concentrations, but further studies are needed to determine the diagnostic accuracy of the measurement of fCal for this purpose. Additional studies are also warranted to evaluate the potential association of fCal concentrations with mucosal calprotectin expression and to determine the potential utility of fCal measurements for treatment monitoring in FCE.

## Figures and Tables

**Figure 1 vetsci-10-00419-f001:**
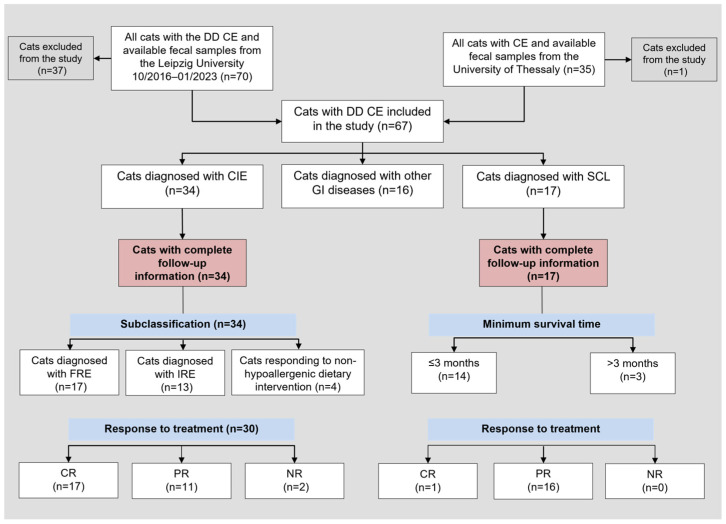
Flowchart of all cats with the differential diagnosis of chronic enteropathy included in the study and those cats with complete follow-up information. Abbreviations: CE = chronic enteropathy; CIE = chronic inflammatory enteropathy; CR = complete remission; DD = differential diagnosis; FRE = food-responsive enteropathy, GI = gastrointestinal; IRE = immunosuppressant-responsive enteropathy; *n* = number (count); NR = no response; SCL = small-cell lymphoma; PR = partial remission. State of remission was based on clinical signs, partial remission in SCL cases was defined as “stable disease”. Classification as PR or CR was not documented for 4 cats in the CIE group.

**Figure 2 vetsci-10-00419-f002:**
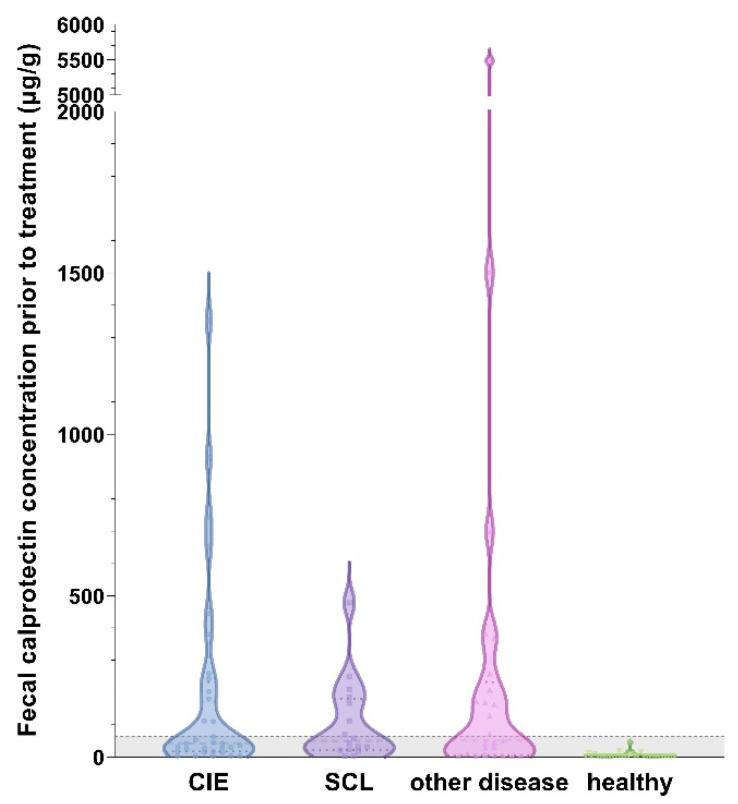
Violin plot showing fecal calprotectin (fCal) concentrations in all cats (*n* = 99) included in the study. All disease groups of cats had significantly higher fCal concentrations than healthy control cats (all *p* < 0.0001), with no differences among cats with chronic inflammatory enteropathy (CIE), small-cell alimentary lymphoma (SCL), or other gastrointestinal (GI) or extra-GI diseases (all *p* > 0.05). A total of 22 cats in the CIE group, 10 cats in the SCL group, 7 cats in the diseased control group, and all of the healthy controls had fCal levels within the reference interval. Gray-shaded area below the horizontal dashed line represent the reference interval (<64 µg/g) [43]. Note the broken y-axis.

**Figure 3 vetsci-10-00419-f003:**
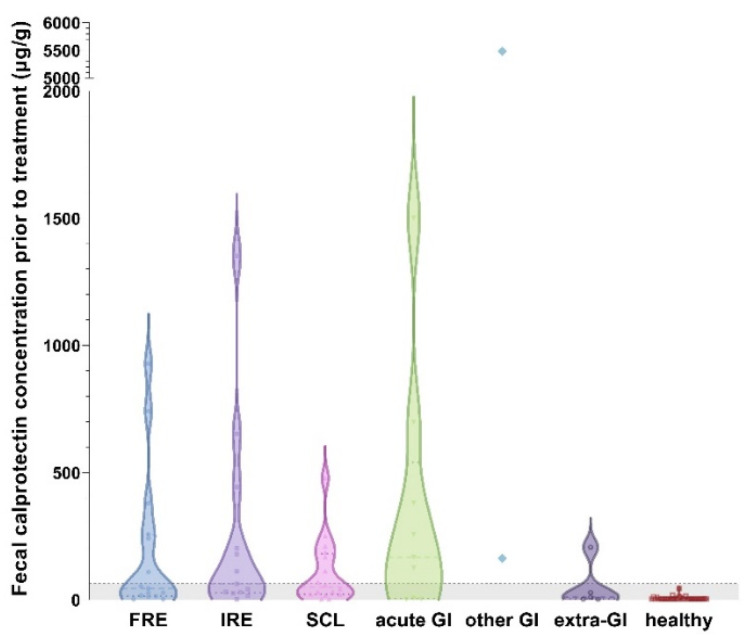
Violin plot showing fecal calprotectin (fCal) concentrations in all 7 groups of cats enrolled in the study. Fecal calprotectin concentrations did not distinguish cases of food-responsive enteropathy (FRE; median: 45 µg/g, IQR: 14–258 µg/g) from immunosuppressant-responsive enteropathy (IRE; median: 63 µg/g, IQR: 28–324 µg/g; *p* = 0.5189) or small-cell intestinal lymphoma (SCL; median: 50 µg/g, IQR: 21–181 µg/g; *p* = 0.9842). Fecal calprotectin concentrations were highest in cats with acute gastrointestinal (GI) disease (median: 168 µg/g, IQR: 6–540 µg/g), with no significant difference compared to any chronic enteropathy group (all *p* > 0.05). Cats with extra-GI disease had the lowest fCal concentrations (median: 10 µg/g, IQR: 3–118 µg/g), showing no difference from healthy controls (*p* = 0.6919). Cats with other GI diseases (e.g., neoplasia) could not be included in the statistical analysis due to the small group size. Gray-shaded area below horizontal dashed line: reference interval (<64 µg/g) [43]. Note the broken y-axis.

**Figure 4 vetsci-10-00419-f004:**
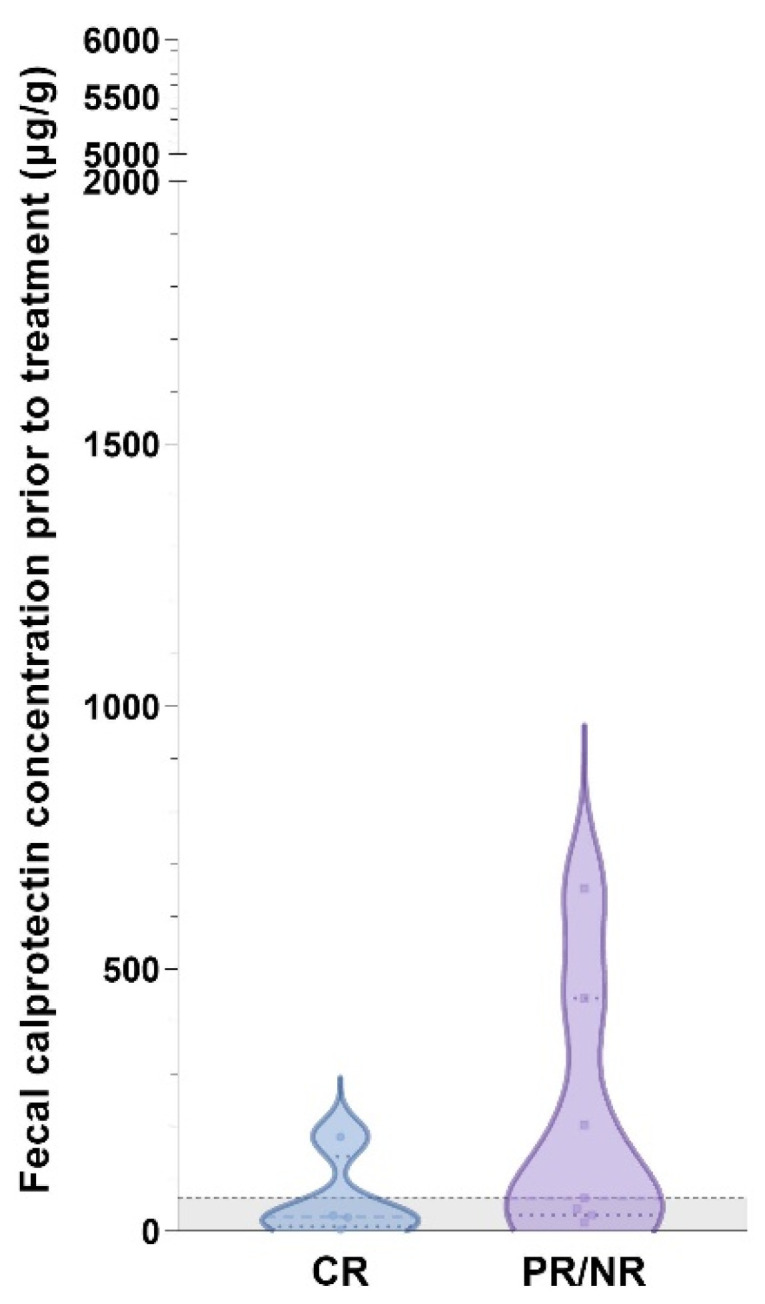
Violin plot showing fecal calprotectin concentrations in cats with IRE and complete follow-up (*n* = 11) included in the study. Fecal calprotectin concentrations were numerically higher in cats with IRE and partial or no response (PR/NR) to treatment (median: 63 µg/g, IQR: 31–444 µg/g) compared to those cats with complete clinical remission (CR; median: 28 µg/g, IQR: 8–143 µg/g) but the difference was not statistically significant (*p* = 0.1564). Gray-shaded area below horizontal dashed line: reference interval (<64 µg/g) [43].

**Table 1 vetsci-10-00419-t001:** Patient data in cats with chronic inflammatory enteropathy (CIE; *n* = 34), small-cell alimentary lymphoma (SCL; *n* = 17), disease controls (*n* = 16), and healthy controls (*n* = 32).

Patient Characteristic	CIE	SCL	Disease Controls	Healthy Controls	*p*-Value
Age in years, median (IQR)	**9.0 (4.8–11.3) ^A^**	**13.0 (11.0–14.0) ^B^**	9.0 (2.0–13.0) ^A,C^	**5.5 (2.0–10.0) ^C^**	**0.0004**
Sex, male (neutered)/female (spayed)	24 (22)/10 (10)	10 (8)/7 (7)	9 (9)/7 (5)	19 (18)/13 (12)	0.6887
Body weight in kg, median (IQR)	4.4 (3.6–5.7) *	5.5 (3.8–6.0)	4.0 (3.3–4.9) ^†^	4.5 (4.2–5.5) ^‡^	0.2099
BCS, median (IQR)	5.0 (3.5–5.5) ^$^	6.0 (4.0–7.0) ^§^	5.5 (3.5–7.5) ^¶^	5.5 (4.5–5.5) ^$^	0.4784
Breed, *n* (%) - Domestic (European) Shorthair - other breeds	24 (71%) 10 (29%)	14 (82%) 3 (18%)	9 (56%) 7 (44%)	26 (81%) 6 (19%)	0.2455
Negative retrovirus (FeLV/FIV) status	8 (100%) **	2 (100%) ^††^	5 (100%) ^‡‡^	–	–
Clinical signs present in months, median (IQR)	8.0 (1.0–24.0) ^§^	0.6 (0.2–1.0) ^††^	1.0 (1.0–12.0) ^$$^	–	0.1413
Minimum survival time in months, median (IQR)	3.0 (3.0–12.0) ^§§,A,B^	**3.0 (3.0–3.0) ^¶¶,A^**	**9.0 (3.5–18.0) ^†,B^**	–	**0.0362**
Number of sites biopsied, median (IQR)	4 (4) ^#^	2 (2) ^¶¶^	3 (2–4) ^††^	–	
** *Clinical parameters* **					
FCEAI score, median (IQR) - severity of reduced activity - severity of vomiting - severity of diarrhea - severity of weight loss - severity of hyporexia	**6 (3–9) ^A^** 1 (0–2) ^A^ **1 (0–2) ^A^** **1 (0–2) ^A^** 1 (0–1.5) ^A^ 1 (0–2) ^A^	**10 (7–11.5) ^B^** 1 (0–2) ^A^ **2 (1–3) ^B^** **2 (1–3) ^B^** 1 (0.5–2) ^A^ 1 (0.5–2.5) ^A^	**5.5 (2.5–7) ^A^** 1 (0–2) ^§,A^ **1 (0–2) ^§,A^** **1.5 (0–2) ^A^** 0 (0–2) ^§,A^ 0 (0–2) ^§,A^	**0 (0)** ^C^ 0 (0) ^B^ 0 (0) ^C^ 0 (0) ^C^ 0 (0) ^B^ 0 (0) ^B^	**<0.0001** **<0.0001** **<0.0001** **<0.0001** **<0.0001** **<0.0001**
Presence of endoscopic lesions, *n* (%) - stomach ^!^ - duodenum/jejunum ^!^ - ileum ** - colon **	27 (100%) ^#^ - 3/7 (43%) - 6/7 (86%) - 5/7 (71%) - 3/6 (50%)	17 (100%) - 1/2 (50%) - 2/2 (100%) – - 1/1 (100%)	2 (100%) ^††^ - 1/2 (50%) - 2/2 (100%) - 0/1 (0%) - 1/1 (100%)	–	– 0.9742 0.6187 0.1372 0.3219
Presence of dermatological signs, *n* (%) ^$$^	3/8 (38%)	0/2 (0%)	3/4 (75%)	–	0.2112
Stress association of clinical signs, *n* (%) ^¶¶^	6/8 (75%) ^A,B^	**0/2 (0%) ^A^**	**7/7 (100%) ^B^**	–	**0.0084**
** *Clinicopathologic parameters* **					
Serum cobalamin in ng/L, median (IQR)	**914 (430–1001) ^A,&^**	**318 (178–637) ^B^**	355 (197–963) ^A,B,^**	–	**0.0054**
Hypocobalaminemia, *n* (%)	4/30 (13%) ^&^	7/17 (42%)	2/8 (25%) **	–	0.0717
Serum folate in µg/L, median (IQR)	18.1 (13.0–22.8) ^#^	16.3 (8.4–30.8) ^^^	14.1 (11.9–29.3) **	–	0.7887
Hypofolatemia, *n* (%)	3/28 (11%)	3/16 (19%)	0/8 (0%)	–	0.2615
Hyperfolatemia, *n* (%)	5/28 (18%)	5/16 (31%)	2/8 (25%)	–	0.5967
Serum total protein in g/L, median (IQR)	73 (66–79) ^ß^	72 (62–74) ^§^	75 (70–81) ^§^	–	0.2925
Hypoproteinemia, *n* (%)	2/32 (6%)	3/15 (20%)	0/12 (0%)	–	0.0950
Hyperproteinemia, *n* (%)	3/32 (9%)	1/15 (7%)	4/12 (33%)	–	0.1167
Serum albumin in g/L, median (IQR)	**36 (33–40) ^A,&^**	**36 (30.5–40) ^A,^^**	**31 (26.5–34.5) ^B,$$^**	–	**0.0222**
Hypoalbuminemia, *n* (%)	4/30 (13%)	2/16 (13%)	2/14 (14%)	–	0.9898
Serum globulin in g/L, median (IQR)	**33.5 (31–43.5) ^A,&^**	**32 (30–36) ^A,§^**	**42 (36.5–51) ^B,%^**	–	**0.0181**
Hyperglobulinemia, *n* (%)	10/30 (33%) ^A,B^	**3/15 (20%) ^A^**	**8/12 (67%) ^B^**	–	**0.0379**
Serum total calcium in mmol/L, median (IQR)	2.50 (2.40–2.63) ^#^	2.54 (2.33–2.69) ^^^	2.50 (2.40–2.60) ^##^	–	0.8709
Total hypocalcemia, *n* (%)	0/27 (0%)	2/16 (13%)	1/7 (14%)	–	0.0864
Serum BUN in mmol/L, median (IQR)	**18 (8–22.5) ^A,!!^**	**23 (19.5–30) ^B,^^**	**9 (7.5–12.5) ^C,&&^**	–	**0.0001**
Serum BUN increase, *n* (%)	**18/29 (62%) ^A^**	**15/16 (94%) ^B^**	4/13 (31%) ^A^	–	**0.0010**
Serum phosphorus in mmol/L, median (IQR)	0.47 (0.34–0.56) ^#^	0.52 (0.44–0.54) ^^^	0.61 (0.45–0.65) ^##^	–	0.1108
Hypophosphatemia, *n* (%)	2/28 (7%)	0/16 (0%)	0/7 (0%)	–	0.2915
Serum ALT activity in U/L, median (IQR)	59 (37–88) ^&^	68 (52–121) ^^^	65 (47–156) ^^^^	–	0.3389
Increased serum ALT activity, *n* (%)	8/31 (26%)	7/16 (44%)	2/9 (22%)	–	0.3898
Serum ALP activity in U/L, median (IQR)	32 (25–48) ^§§^	32 (26–51) ^^^	56 (29–70) ^&&^	–	0.4767
Increased serum ALP activity, *n* (%)	5/32 (16%)	2/16 (13%)	5/14 (36%)	–	0.2368
Serum tT4 in nmol/L, median (IQR)	**24.5 (20.8–34.3) ^A,ßß^**	**34.5 (29.5–37.6) ^B,&&^**	43.2 (21.9–50.2) ^A,B,##^	–	**0.0145**
Serum Spec fPL in μg/L, median (IQR)	2.0 (1.4–4.2) ^&^	2.7 (2.5–3.3) ^¶¶^	1.7 (1.4–4.2) **		0.0752
Increased serum Spec fPL, *n* (%)	8/31 (26%)	4/17 (24%)	2/8 (25%)	–	0.2368
Serum fTLI in μg/L, median (IQR)	**21.4 (15.7–35.3) ^A,%%^**	**60.5 (36.3–69.8) ^B,$$^**	45.7 (31.6–102.4) ^A,B,###^	–	**0.0024**
Serum fructosamine in μmol/L, median (IQR)	225 (203–277) ^^^^	–	254 (232–286) ^¶^	–	0.5161
** *Sonographic abnormalities* **					
Increased gastrointestinal wall thickness, *n* (%) ^!!!^	**12/13 (92%) ^A^**	**2/2 (100%) ^A^**	**4/11 (36%) ^B^**	–	**0.0049**
Thickened tunica muscularis layer, *n* (%) ^!!!^	7/13 (54%)	1/2 (50%)	4/11 (36%)	–	0.6867
Loss of gastrointestinal wall layering, *n* (%) ^!!!^	3/13 (23%)	1/2 (50%)	0/11 (0%)	–	0.0637
Enlarged regional lymph nodes, *n* (%) ^!!!^	11/13 (85%)	1/2 (50%)	5/11 (46%)	–	0.1081
Evidence of free abdominal fluid, *n* (%) ^!!!^	5/13 (39%)	0/2 (0%)	3/11 (27%)	–	0.3902
** *Fecal biomarker of inflammation* **					
Fecal calprotectin in μg/g, median (IQR)	**42 (17–232) ^A^**	**50 (21–181) ^A^**	**145 (5–351) ^A^**	3 (3) ^B^	**<0.0001**

BCS: body condition score (scale from 1–9); FCEAI: feline chronic enteropathy activity index; IQR: interquartile range. Parameters in bold font and differential superscript capital letters indicate significant differences at *p* < 0.05. * recorded for *n* = 33 cats; ^†^ recorded for *n* = 13 cats; ^‡^ recorded for *n* = 25 cats; ^$^ recorded for *n* = 26 cats; ^§^ recorded for *n* = 15 cats; ^¶^ recorded for *n* = 6 cats; ** recorded for *n* = 8 cats; ^††^ recorded for *n* = 2 cats; ^‡‡^ recorded for *n* = 5 cats; ^$$^ recorded for *n* = 14 cats; ^§§^ recorded for *n* = 31 cats; ^¶¶^ recorded for *n* = 17 cats; ^#^ recorded for *n* = 27 cats; ^!^ recorded for *n* = 11 cats; ^&^ recorded for *n* = 30 cats; ^^^ recorded for *n* = 16 cats; ^ß^ recorded for *n* = 28 cats; ^%^ recorded for *n* = 12 cats; ^##^ recorded for *n* = 7 cats; ^!!^ available from *n* = 29 cats; ^&&^ available from *n* = 13 cats; ^^^^ available from *n* = 9 cats; ^ßß^ available from *n* = 22 cats; ^%%^ available from *n* = 21 cats; ^###^ available from *n* = 3 cats; ^!!!^ available from *n* = 26 cats.

**Table 2 vetsci-10-00419-t002:** Correlation among clinical, laboratory, and histologic findings in cats with CIE. Summarized are the relationships between fecal calprotectin (fCal) concentrations, feline chronic enteropathy activity index (FCEAI) scores, clinicopathologic results (i.e., serum albumin, globulin, and cobalamin concentrations), and the severity of intestinal inflammatory and morphologic histologic lesions in cats with CIE (*n* = 34).

Parameter		Spearman *ρ* Correlation Coefficient (*p*-Value)	
	Correlated with	fCal Concentration	FCEAI Score
** *Clinical criteria* **			
FCEAI score		0.21 (0.2546)	-
** *Serum protein concentrations* **			
Serum albumin concentration		**−0.45 (0.0165)**	−0.23 (0.2135)
Serum globulin concentration		**0.48 (0.0104)**	0.01 (0.9773)
** *Serum functional biomarker* **			
Serum cobalamin concentration		**−0.40 (0.0335)**	**−0.49 (0.0058)**
** *Histologic criteria* **			
Histologic lesions (composite score) ^¶^		0.37 (0.0789)	0.09 (0.6777)
Morphologic criteria ^¶^		0.30 (0.1540)	0.21 (0.3262)
Inflammatory criteria ^¶^		0.23 (0.2949)	0.01 (0.9483)
Duodenum/jejunum (composite score)		**0.44 (0.0305)**	0.39 (0.0526)
Morphologic criteria (sum)		0.29 (0.1761)	0.17 (0.4118)
- Villus stunting		**0.62 (0.0011)**	0.19 (0.3611)
- Epithelial injury		0.28 (0.1824)	**0.40 (0.0486)**
- Crypt distension		**−0.71 (0.0001)**	0.01 (0.9798)
- Lacteal dilation		0.11 (0.6232)	−0.03 (0.8924)
- Mucosal fibrosis		**0.43 (0.0366)**	0.11 (0.5949)
Inflammatory criteria (sum)		0.35 (0.0913)	0.39 (0.0560)
- Intraepithelial lymphocytes		0.22 (0.3141)	−0.02 (0.9184)
- Lamina propria LPC		0.20 (0.3416)	**0.44 (0.0304)**
- Lamina propria eosinophils		n/a	n/a
- Lamina propria neutrophils		0.20 (0.3446)	0.19 (0.3713)
- Lamina propria ΜΦ		0.23 (0.2874)	0.33 (0.1097)
Ileum (composite score)		0.35 (0.1161)	0.23 (0.2993)
Morphologic criteria (sum)		0.32 (0.1570)	0.41 (0.0618)
- Villus stunting		0.34 (0.1374)	0.14 (0.5359)
- Epithelial injury		**0.52 (0.0149)**	0.42 (0.0527)
- Crypt distension		0.37 (0.0983)	0.14 (0.5385)
- Lacteal dilation		n/a	n/a
- Mucosal fibrosis		−0.03 (0.8854)	**0.47 (0.0246)**
Inflammatory criteria (sum)		0.27 (0.2445)	−0.03 (0.9133)
- Intraepithelial lymphocytes		−0.14 (0.5444)	−0.23 (0.2978)
- Lamina propria LPC		0.32 (0.1541)	0.01 (0.9524)
- Lamina propria eosinophils		n/a	n/a
- Lamina propria neutrophils		0.26 (0.2450)	0.34 (0.1125)
- Lamina propria ΜΦ		0.26 (0.2450)	0.34 (0.1125)
Colon (composite score)		−0.11 (0.6280)	−0.22 (0.3080)
Morphologic criteria (sum)		−0.18 (0.4085)	−0.17 (0.4183)
- Epithelial injury		0.36 (0.0823)	−0.11 (0.6174)
- Goblet cell loss or hyperplasia		−0.38 (0.0731)	−0.24 (0.2511)
- Crypt dilation and distortion		−0.13 (0.5266)	−0.14 (0.5002)
- Mucosal fibrosis and atrophy		−0.16 (0.4575)	0.09 (0.6746)
Inflammatory criteria (sum)		0.18 (0.3954)	−0.16 (0.4399)
- Intraepithelial lymphocytes		0.28 (0.1807)	0.07 (0.7440)
- Lamina propria LPC		0.03 (0.9033)	−0.18 (0.3793)
- Lamina propria eosinophils		n/a	n/a
- Lamina propria neutrophils		n/a	n/a
- Lamina propria ΜΦ		0.36 (0.0803)	0.17 (0.3971)

ΜΦ: macrophages; LPC: lymphocytes/plasma cells; n/a: not applicable; blue shaded cells: statistically significant (*p* < 0.05) only without Bonferroni correction; orange-shaded cells: statistical significance (*p* < 0.05) remaining after Bonferroni correction (*n* = 2, 3, 4, or 5); ^¶^ calculated only when duodenum, ileum, and colon were sampled and evaluated.

## Data Availability

Data (anonymized) are available from the first, second, or last author upon reasonable request.

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
