# Peer review of "Fecal Calprotectin Concentrations in Cats with Chronic Enteropathies"

_vetsci, 2023, doi:10.3390/vetsci10070419_

Round 1

Reviewer 1 Report

This is an interesting study, contributing to the literature on feline chronic enteropathies. There are minor questions, which I would ask the authors to address.

Materials and methods:

What histopathologic criteria were used to distinguish inflammatory enteropathies from GI lymphoma? While it is mentioned that IHC was performed in some cats, were there specific criteria which directed IHC performance? As CIE and SCL can be challenging to distinguish based on even histopathology alone, this information would be relevant to provide.

Results:

Lines 248-249: It is unclear why cats with small cell lymphoma + positive Giardia antigen were included in the SCL group, rather than the other GI disease group, when a cat with hyperthyroidism and CIE was included in the other GI disease group.

Reviewer 2 Report

Table 1: thje resolution of Table 1 is of low quality and is difficult to read

Line 464: there is an unclosed parenthesis

I believe that the work is well written, it is an original study and it can be a good starting point to better evaluate the role of fCal also in cat diseases. There are several limitations to the study as also reported by the authors but this does not limit the originality of the study and it can be a good starting point for future studies. References are appropriate. However, it is necessary to improve the quality of the tables as they are difficult to read because the characters are too small and by enlarging the image the quality is lost. 

Reviewer 3 Report

The paper explores the possible use of fecal calprotectin for FCE diagnosis in cats.

The results presentation must be improved. It is difficult to read the tables, and a more concise way should be used to highlight main findings. Part of table 1 is not legible, part of the text was missing.

Although calprotectin is relatively resistant to proteolysis, sample frozen at -20 is not recommended for more than 1 week (https://doi.org/10.1093/ecco-jcc/jjaa093). For how long were samples maintained at -20 prior to analysis?

The disease control group end up having high variability and is difficult to understand if some of the conclusions are statistically supported as each pathology, except acute GI disease, is poorly represented.

Does the values of neutrophils between groups are significantly different? Where results analyzed according to the levels of neutrophils as calprotectin should be directly related to this cell type?

Calprotectin values among animals with CIE and SCL are also very variable. It would be useful to know how many animals in each group still had normal values of calprotectin, according to the value obtained from healthy controls and correlate these subgroups with the other findings.

No comments

Round 2

Reviewer 3 Report

All comments were addressed.

Minor revision is required.